# Biophysical ordering transitions underlie genome 3D re-organization during cricket spermiogenesis

Guillermo A. Orsi [1,7] ✉, Maxime M. C. Tortora [2,6,7], Béatrice Horard[2], Dominique Baas[3], Jean-Philippe Kleman [4], Jonas Bucevičius[5], Gražvydas Lukinavičius [5], Daniel Jost [2] ✉ & Benjamin Loppin [2] ✉

Spermiogenesis is a radical process of differentiation whereby sperm cells acquire a compact and specialized morphology to cope with the constraints of sexual reproduction while preserving their main cargo, an intact copy of the paternal genome. In animals, this often involves the replacement of most histones by sperm-specific nuclear basic proteins (SNBPs). Yet, how the SNBP-structured genome achieves compaction and accommodates shaping remain largely unknown. Here, we exploit confocal, electron and super-resolution microscopy, coupled with polymer modeling to identify the higher-order architecture of sperm chromatin in the needle-shaped nucleus of the emerging model cricket *Gryllus bimaculatus*. Accompanying spermatid differentiation, the SNBP-based genome is strikingly reorganized as ~25nm-thick fibers orderly coiled along the elongated nucleus axis. This chromatin spool is further found to achieve large-scale helical twisting in the final stages of spermiogenesis, favoring its ultracompaction. We reveal that these dramatic transitions may be recapitulated by a surprisingly simple biophysical principle based on a nucleated rigidification of chromatin linked to the histone-to-SNBP transition within a confined nuclear space. Our work highlights a unique, liquid crystal-like mode of higher-order genome organization in ultracompact cricket sperm, and establishes a multidisciplinary methodological framework to explore the diversity of non-canonical modes of DNA organization.

During spermiogenesis, post-meiotic spermatogenic cells differentiate to fulfill their main function of transporting an intact paternal genome into the egg. This genome must fit into a cell that is morphologically optimized for the specific requirements of sexual reproduction in every species[1–3]. Indeed, there is a remarkable diversity in the morphology of sperm nuclei in animals, including. falciform-shaped (mouse), rod-shaped (rooster), drill-bit-shaped (house sparrow), or the needle-shaped sperm nucleus that is common among insects[2]. Fitting into these complex morphologies, the paternal nucleus typically undergoes a high level of compaction – ~10x to ~40x compared to

[1]Institute for Advanced Biosciences, University Grenoble Alpes, Inserm U 1209, CNRS UMR 5309, 38000 Grenoble, France. [2]Laboratoire de Biologie et Modélisation de la Cellule, École Normale Supérieure de Lyon, CNRS UMR5239, Inserm U1293, Université Claude Bernard Lyon 1, Lyon, France. [3]Laboratoire MeLiS, CNRS UMR 52684, Inserm U 1314, Institut NeuroMyoGène, Université Claude Bernard Lyon 1, Lyon, France. [4]Institut de Biologie Structurale, UMR5075, University Grenoble Alpes, CEA, CNRS, Grenoble, France. [5]Chromatin Labeling and Imaging Group, Department of NanoBiophotonics, Max Planck Institute for Multidisciplinary Sciences, Göttingen, Germany. [6]Present address: Department of Quantitative and Computational Biology, University of Southern California, Los Angeles, CA, USA. [7]These authors contributed equally: Guillermo A. Orsi, Maxime M. C. Tortora. ✉e-mail: guillermo.orsi@inserm.fr; daniel.jost@ens-lyon.fr; benjamin.loppin@ens-lyon.fr

somatic interphase cell nuclei, depending on the species[4,5]. Abnormal compaction and/or morphology are generally associated with reduced fertility[6–8], illustrating the physiological importance of these mechanisms.

Differentiating, post-meiotic male germ cells (spermatids) undergo re-shaping accompanied by removal of excess cytoplasm and formation of sperm-specific structures typically including the apical acrosome and the basally anchored sperm flagellum[9,10]. Actin filaments and microtubules play a critical role in this remodeling process[9,10]. The sperm head is at least in part shaped by a stack of F-actin hoops anchored at the acrosome-nucleus junction (a structure called acroplaxome) and a transient microtubule-actin basket-like structure known as the manchette, that encircles the nucleus. These structures are thought to act as scaffolding fibers contributing to constrained spermatogenic cell elongation by contractile forces. Additionally, as the manchette forms, nuclear pores are redistributed caudally and are no longer found in fully mature spermatozoa[11]. Concomitantly, the importance of multiple sperm-specific lamin, lamin-associated and LINC (linker of nucleoskeleton and cytoskeleton) proteins for sperm nuclear apico-basal polarization, shaping and compaction is well-documented[12,13].

In contrast, the link between sperm nuclear morphology establishment and genome tridimensional reorganization has remained unclear. At the chromatin level, spermiogenesis in many animals involves widespread remodeling[14–16]. In most eukaryotic cells, chromatin is organized as a string of nucleosomes, composed of ~146 bp of DNA wrapped around an octamer of histone proteins. Histone proteins can further undergo post-translational modifications that alter their properties and enable their interaction with effector complexes[17]. While nucleosomes are essential to organize chromatin and modulate its functions in somatic nuclei, the vast majority of histones are removed from chromatin during sperm differentiation and replaced by Sperm-specific Nuclear Basic Proteins (SNBPs) in many animal species, including mammals and insects[14–16]. In mouse and *Drosophila*, histones first become broadly acetylated[18–20], a modification that entails changes in the biophysical properties of the chromatin, favoring DNA accessibility and increasing short-range rigidity[21]. Acetylated histones are further recognized by testis-specific bromodomain proteins (BRDTs)[22,23]. Via BRDTs, acetylated histones are evicted and replaced on DNA by Transition Proteins (TPs) and Protamines in mammals or Transition protein-like (TPL) and Protamine-like proteins (PLs) in Drosophila. These SNBPs are highly basic, fast-evolving small proteins[24,25]. In addition, Protamines and PLs usually feature multiple cysteine residues, which can form intra- and/or inter-molecular covalent disulfide bonds that participate in Protamines and PLs stability on DNA[26–28]. Disulfide bond reduction at fertilization is thus required for paternal chromatin decompaction, PL removal, in preparation of nucleosome assembly and paternal genome incorporation into the first zygotic nucleus[29]. In vitro, mammalian Protamines acting as multivalent cations can organize DNA into ~150nm-wide tori[24,30–32]. Extracted chromatin from vertebrate sperm confirmed the presence of torus- as well as rod-shaped structures[33]. Yet, the actual in situ configuration of the genome in mature sperm is notoriously difficult to access, in large part due to the extreme density and insolubility of this substrate.

Important insights into this issue came from electron microscopy studies of insect spermatogenesis carried out over the last seventy years, which revealed an outstanding diversity in the large-scale re-configuration of the nuclear content[34]. In insects, although sperm chromatin is generally diffuse in spermatocytes (meiotic germ cells) and highly compact in mature sperm, genome organization in intermediate spermatid stages can widely vary between species[35]. A frequently observed feature is the progressive thickening of chromatin fibers, sometimes further organizing into lamellae, which have been proposed to arise from liquid-liquid phase separation (via either spinodal decomposition or nucleation and growth)[35]. Patterned chromatin condensation has notably been observed during spermiogenesis in several orthopteran insects including the grasshopper *Chortophaga viridifasciata*[36], the cave cricket *Ceutophilus nigricans*[37] or the bush katydid *Scudderia*[38]. As a striking case in point, pioneering studies on the two-spotted cricket *Gryllus bimaculatus* revealed a highly regular configuration of DNA in maturing sperm, consisting of ~30nm-thick chromatin fibers that were bundled, stretched, and coiled along the needle-shaped nucleus axis[39]. This remarkable organization is arguably among the simplest solutions to higher-order genome folding in any cell type or species, thus providing a unique opportunity to dissect the mechanisms underlying large-scale remodeling of nuclear architecture.

Here, we investigate the underlying principles of such drastic spatial rearrangements of the genome in *G. bimaculatus* male germ cells. By exploiting confocal, electron and single-molecule localization microscopy, we identify the successive chromatin structural and architectural transitions that accompany spermiogenesis. To understand their mechanistic, physical bases, we establish a simple polymer modeling approach to simulate genome dynamics in the restricted sperm nuclear space. By iteratively confronting simulations and observations, we reveal that chromatin spooling can be explained by a nucleated, gradual increase in chromatin rigidity during the histone-to-protamine transition. We further uncover that the alignment of spooled fibers along the nuclear axis does not precede, but rather follows antero-posterior cell polarization and elongation, indicating that chromatin configuration may adapt to cell shape in a liquid crystal-like fashion. Finally, we provide evidence that spooled fibers twist around the nucleus axis during terminal differentiation to achieve an ultracompact configuration. Our results demonstrate how a relatively simple set of changes in the biophysical properties of the chromatin fiber may lead to radical tridimensional reorganization of the genome that allows its fitting into the sperm nucleus.

## Results
### Ordered coil configuration of chromatin fibers in sperm following the histone-to-protamine transition
We first sought to better characterize spermiogenesis in *G. bimaculatus*, i.e. the differentiation of haploid post-meiotic germ cells. Based on nuclear morphology, by analogy to well-characterized spermiogenesis stages in *Drosophila*[14], we identified multiple steps in germ cell proliferation and differentiation including spermatocytes, round spermatids, canoe (elongating) spermatids, needle-shaped spermatids (Fig. 1A), as well as mature spermatozoa. As previously reported, the needle-shaped nuclei in mature cells were ~0,5 μm in width and ~20 μm in length[39]. This same study has shown that histones are lost in mature sperm nuclei, where only a single species of acid-extractable SNBP remains[39]. Accordingly, with immunofluorescence using an antibody that broadly recognizes histone core domains and linker histone H1, we confirmed that the strong histone signal in early elongating spermatids is undetectable at later stages (Fig. 1A, B). Consistently, histone H3 could be readily revealed by Western blot on testes extracts but was undetectable in mature sperm extracts (Fig. 1C). These results confirm a global histone-to-SNBP replacement in this species.

We further corroborated by electron microscopy the remarkable organization of chromatin into bundled, coiled fibers (Fig. 1Dl, a). We found that ~25nm-thick fibers stretch into ~200 linear segments along the nucleus axis in canoe spermatids (Fig. 1Dt). We observed that the flagellum cytoskeletal structures are lodged within an indentation at the basal-most end of the nucleus (Fig. 1Db). At the apical – acrosomal – end, we detected bending of the fibers, as previously described (Fig. 1Da)[39]. Sagittal sections of these ~25 nm-thick fibers (Fig. 1Ds) further revealed an internal structure in which DNA appears to be

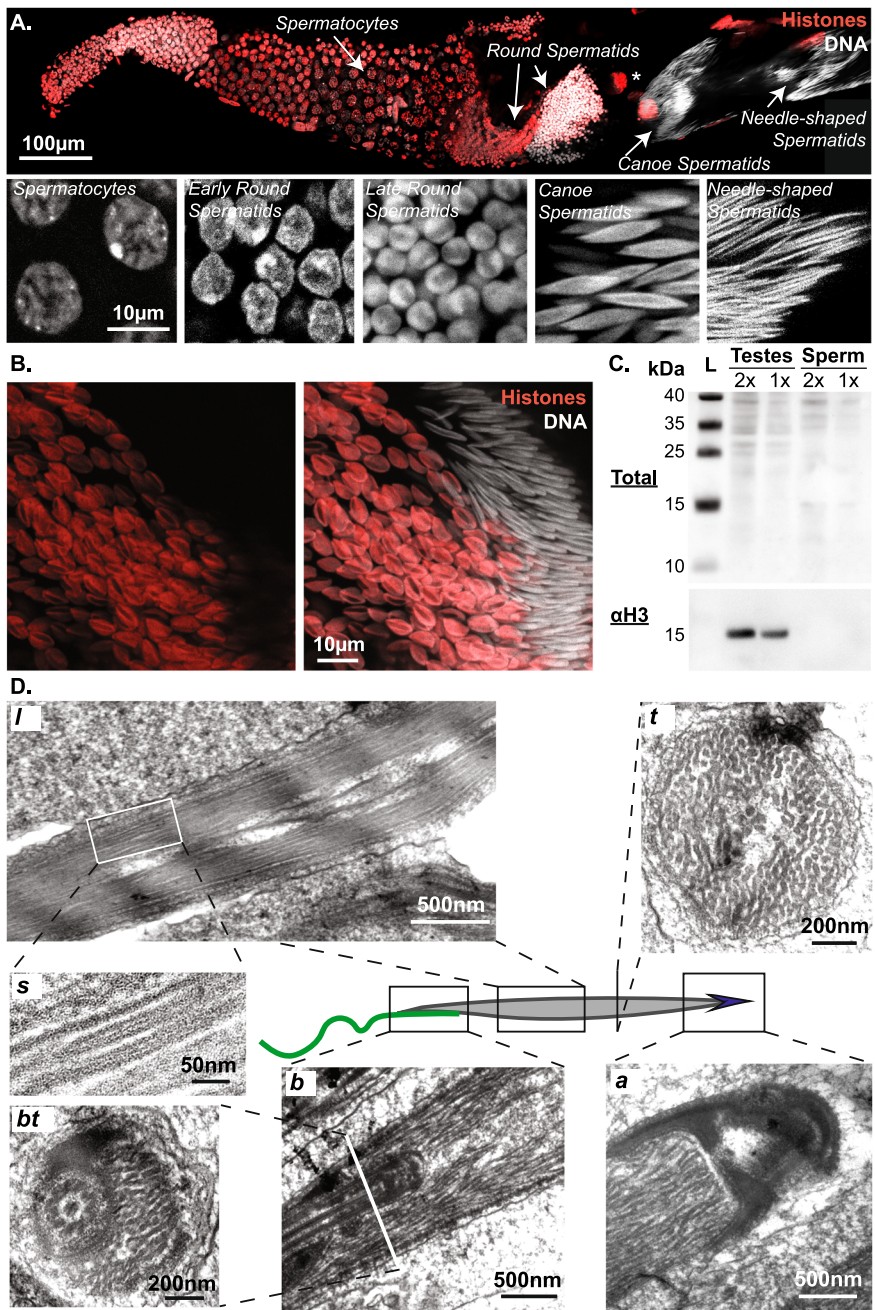

**Fig. 1 | Coiled organization of chromatin fibers in *Gryllus bimaculatus* sperm.**
**A** Overview of spermiogenesis in *Gryllus bimaculatus*. Confocal microscopy image of a single follicle from an adult testis stained for DNA (white) and histones (red). Multiple cysts are visible in which male germ cells differentiate synchronously. From our estimations, the premeiotic cysts contain 512 primary spermatocytes and post-meiotic cysts must then contain about 2048 spermatids. By analogy to *Drosophila*, we distinguish successive stages in spermiogenesis including spermatocytes, round spermatids, canoe (elongating) spermatids, and needle-shaped spermatids. *: a polyploid somatic nucleus. **B**, **C** Sperm chromatin undergoes a Histone-SNBP transition. Confocal image showing early elongating (left side) and late elongating (right side) spermatids stained for DNA (white) and histones (red) confirms massive histone eviction during spermatid elongation (**B**). Western blot of protein extracts from testes and sperm revealing whole proteins and histone H3 confirms undetectable histone levels in mature gametes (**C**). L: ladder, 2x 1x: two-fold dilution of protein extract. **D** Chromatin organizes as coiled fibers in spermatids. Electron microscopy images show different sections of spermatids around the canoe stage. A longitudinal section (*l*) of the spermatid nucleus shows chromatin fibers in alignment with the antero-posterior axis. A sagittal section (*s*) of individual chromatin fibers reveals a tubular internal structure. At the basal end (*b*, *bt*), an overlap between the nucleus and flagellum is observed. Note that images in (*b*) and (*bt*) represent longitudinal and transversal sections of this structure. Bending of chromatin fibers is observed at the apical end (*a*). Approximately 200 individualized fibers are distinguished in a transversal section (*t*).

condensed within a protein cylinder featuring a central protein axis. Considering a genome of 1.66 Gb in size[40], we estimated a DNA linear density of ~415 bp/nm along the fiber. With the volume of one base pair being of ~1 nm³, the intra-fiber volume fraction of DNA is close to 100%, compared to ~10% for nucleosomal chromatin fibers. Furthermore, assuming a 500 nm thick and 20 μm long mature sperm nucleus, we evaluated that these fibers occupy ~70% of the nuclear volume, while chromatin occupies ~10% of the nuclear volume in typical somatic nuclei. Together, these observations confirm the remarkable organization of the *G. bimaculatus* sperm genome into an elongated, ordered coil of thick fibers that compact DNA to high levels and occupy most of the nuclear space.

## Stiffening of a polymer in a confined space recapitulates chromatin nuclear re-localization

To understand the mechanical underpinnings of this striking transition, we developed a simple polymer model considering the specific features of sperm chromatin (see "Methods" section). We modeled chromatin as a polymer consisting of identical 30 nm monomers, whose kinetics are controlled by the interplay between thermal motion, steric repulsion, and bending rigidity. This long chain was confined within a flexible and discrete nuclear envelope, modeled as a polymerized elastic shell. Positively charged SNBPs are expected to result in a higher stiffness of sperm chromatin compared to that of nucleosomal chromatin[21,27,41], which we translated into an increase in the local bending rigidity of the polymer. Such rigidity is characterized by the persistence length parameter $l_p$, that depicts the typical length scale at which two bonds along the polymer chain become decorrelated, i.e., informally, the length below which segments of polymer behave as rigid rods. Thus, we represented the mature chromatin fiber as a worm-like chain with $l_p \cong 1.5\ \mu m$, qualitatively consistent with our microscopy imaging data in elongating spermatids. This value should be contrasted with the persistence length $l_p \cong 50\ nm$ of naked DNA, as well as that $l_p \geq 80\ \mu m$ of rigid cytoskeletal filaments such as microtubules[42]. We further modeled the chromatin backbone connectivity via harmonic springs whose finite stiffness was adjusted to emulate a regime of moderate-to-high topoisomerase activity, in order to allow for large-scale rearrangements of the confined chain within computationally-accessible timeframes (see "Methods" section).

First, we simulated the re-organization of a polymer upon a progressive, uniform increase in bending rigidity, starting from an isotropic, disordered conformation of an initially flexible chain. In these conditions, multiple segments of the polymer progressively coiled in different directions (Fig. 2A, left) and preferentially localized at the periphery of the nuclear space upon full rigidification (Fig. 2A, middle). To evaluate the relevance of these predictions, we analyzed early stages in cricket spermatogenesis by electron microscopy. As expected, round ~10 μm diameter nuclei (consistent with that of spermatocytes) displayed a uniform chromatin arrangement: the nuclear content was overall homogeneous, with the exception of a heterochromatin-like area located at the nuclear periphery (Fig. 2B). Upon progression towards the round spermatid stage, we found that round spermatids displayed a ~20x smaller nuclear volume compared to spermatocytes (Figs. 1A and 2B, C). Concomitantly, fluorescence and electron microscopy revealed an increasingly inhomogeneous distribution of nuclear material, characterized by a marked enrichment of chromatin at the nuclear periphery (Fig. 2C, D, F and Supp. Fig. 1). As confirmed by our simulations, this loss of uniformity is characteristic of a strongly confined, stiff polymer, and becomes increasingly pronounced as the polymer persistence length increases relative to the radius of the confining cavity (Fig. 2A, E, F). Hence, the observed peripheral enrichment of chromatin may be attributed to the combination of the rigidification of the fiber with the reduction in nuclear dimensions measured at the round spermatid stage, consistent with the results of our model predictions.

To understand how this inhomogeneously oriented coiled polymer could fill the elongating nucleus during spermiogenesis, we imposed an arbitrary stretching of the envelope through an external radial force applied at two opposite poles (Fig. 2A, right). Upon stretching, the polymer coils persisted but did not align to the elongation axis. These predictions did not fit our observations in electron microscopy whereby early elongating nuclei instead already displayed a pronounced antero-posterior alignment of chromatin fibers (Fig. 2D and Supp. Fig. 1). We conclude that a simple uniform polymer rigidification scenario only partially recapitulates the chromatin reorganization kinetics in vivo.

## Spooled fiber formation is consistent with a nucleated increase in chromatin rigidity

To refine our model, we considered the possibility that changes in polymer properties did not occur simultaneously over the entire genome, but were instead nucleated at discrete regions and spread gradually in *cis* along the fiber (see "Methods" section). In particular, we simulated the dynamical reorganization of chromatin when the local bending rigidity was first increased in a progressive fashion at a limited number of nucleation sites, initially located near the center of the nucleus (Fig. 3A, B). Upon full rigidification of the nucleation domains, this gradual stiffening kinetics was in turn propagated to their immediate adjacent regions along the chain, and the process was repeated iteratively until the entire polymer reached a homogeneous rigidity. Under these assumptions, locally stiffened chromosomal segments were found to be rapidly deported to the nuclear periphery (Fig. 3C), where they adopted a uniformly oriented coiled configuration (Fig. 3A, B and Supp. Fig. 2). Upon completion of the rigidification of the entire fiber, the polymer adopted a cylindrically-symmetric, spool-like overall structure, compatible with the peripheral localization of chromatin within the nucleus visualized in electron micrographs (Fig. 2F).

At intermediate stages the more flexible, yet-unmodified regions were localized within the cylindrical core of this spooled arrangement of stiffened fiber segments, and thus stretched across the center of the nucleus with isotropic organization. This original ordering pattern may be described in terms of a local alignment parameter $\alpha(r)$, which represents the radially averaged degree of local orientational (or nematic) order. As the rigidified fraction of the fiber grows, $\alpha(r)$ displays marked peaks of increasing width near the nuclear envelope walls, in a manner reminiscent of the surface-induced orientational wetting phenomenon already reported in strongly confined polymer solutions (Supp. Figure 2B)[43,44]. To evaluate whether this simulation fatefully recapitulated fiber spooling in vivo, we computed the radially averaged degree of local nematic order $\langle\alpha(r)\rangle$ that we can quantitatively compare with the 2D nematic order parameter computed from a statistical ensemble of electron micrographs (Fig. 3D and Supp. Fig. 1, see "Methods" section). This analysis confirms that the low values of $\langle\alpha(r)\rangle$ observed in simulations prior to fiber rigidification are consistent with the negligible degree of nematic order observed in spermatocytes, while the larger values of $\langle\alpha(r)\rangle$ in the ordered spool agrees with the strong local nematic order measured in the longitudinal sections of elongated spermatids (Fig. 3D, purple and gray bars).

Since the nucleation and spreading of distinctive chromatin properties can notably occur via histone post-translational modifications, we next analyzed the spatio-temporal pattern of histone H4 acetylation (H4ac). H4ac accompanies and is required for the histone-to-protamine transition in other species, including *Drosophila* and mice. In these model species, H4 becomes broadly acetylated throughout the nucleus. In crickets, confocal imaging using an antibody that recognized multiple acetylated forms of H4 revealed a heterogeneous pattern (Fig. 4). H4ac was only detected at small, discrete foci in early round spermatids, consistent with the deposition of this mark at restricted segments in the genome. In early round spermatids, H4ac signal became clustered at the center of the nucleus, consistent with the early stages of nucleated stiffening in our simulations (Figs. 3A, 4). In later-stage round spermatids, we observed a remarkable pattern, whereby the H4ac signal stretched directionally across two poles of the nucleus. These poles were enriched for Histone signal, while DNA staining was in contrast stronger in the complementary nuclear space surrounding H4ac/Histones (Fig. 4). This particular configuration is strikingly similar to intermediate stages predicted by our polymer simulations (Fig. 4). Finally, at early canoe stages, H4ac and Histone stainings weakened and were no longer detected at later stages. Our observations thus support a model in which histone-to-SNBP exchange occurs via a nucleation-and-spreading mechanism

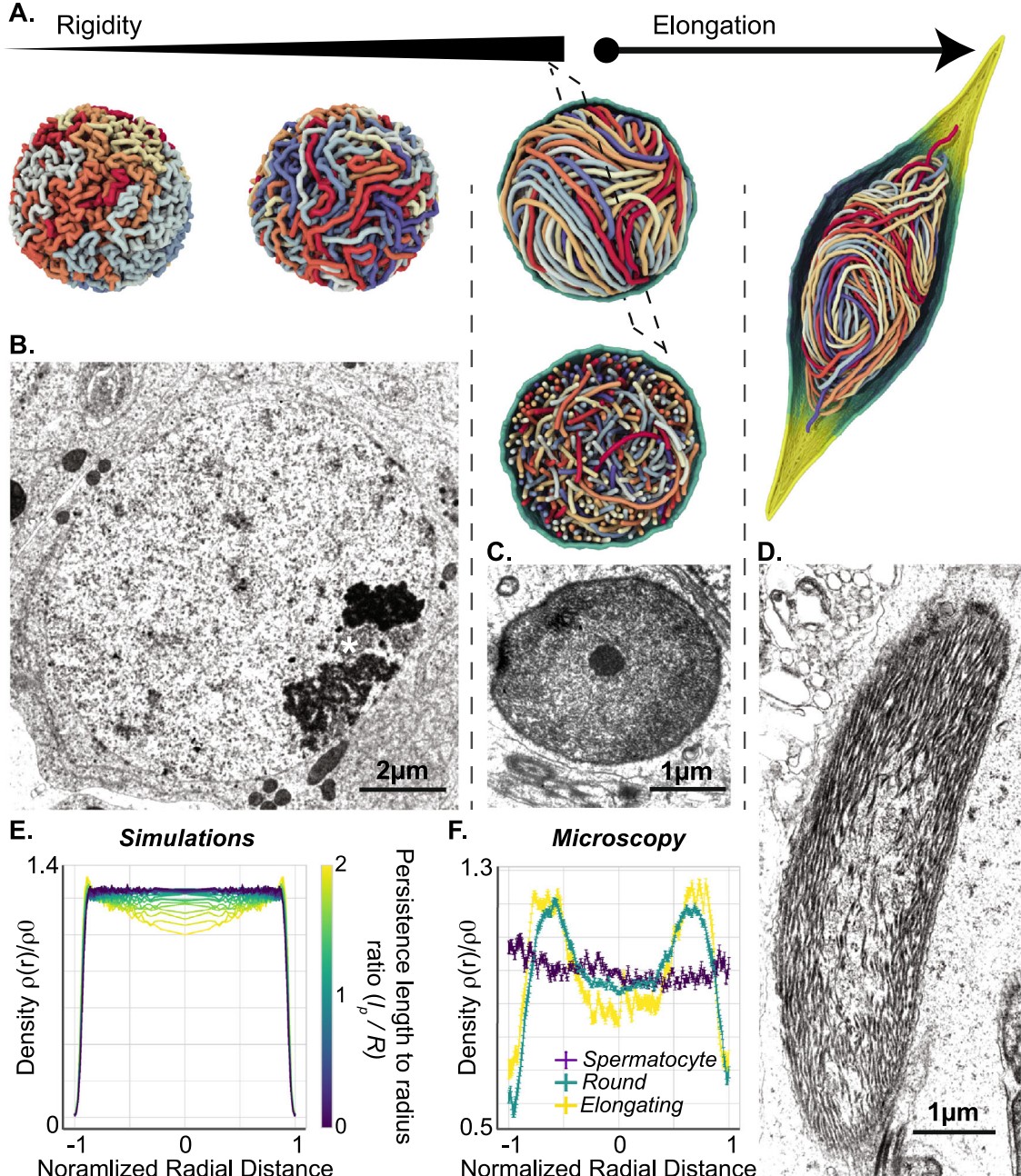

**Fig. 2 | Polymer modeling of rigid fiber dynamics recapitulates features of chromatin localization in sperm.** **A** Chromatin re-organization via a uniform, homogeneous increase in local fiber rigidity in simulated models. Polymer simulations of the stiffening process resulting in a yarn-ball-like state, which is relatively disordered and prevents the establishment of a coherent, homogeneous direction of fiber alignment upon the onset of nuclear elongation. **B**–**D** Chromatin re-organization during spermiogenesis in vivo. Electron microscopy images of spermatocytes (**B**), round (**C**), and elongating (**D**) spermatids show progressive reorganization of chromatin by fiber individualization and alignment. **E** Simulated radial monomer density profiles, normalized by the mean nuclear fiber concentration $\rho_0$, for different persistence-length-to-radius ratios $l_p/R$. Distance is normalized to nucleus radius. Note the growing fiber accumulation at the nuclear periphery as rigidification progresses and $l_p/R$ increases. **F** Experimental density profiles, as computed from TEM images of spermatocytes (purple), round spermatids (green), and elongating spermatids (yellow)(see Methods). Distance is normalized to nucleus radius. Error bars denote the standard error of the mean, and were evaluated using $\mathcal{O}(10)$ independent micrographs taken at the spermatocyte (**B**), round (**C**), and elongating (**D**) developmental stages. In **B**, ' * ' indicates a heterochromatin mass.

that entails chromatin stiffening, and results in a uniformly coiled fiber organization.

The strong local alignment of the spool-like folded arrangement that arises from this rigidification process (Fig. 3D and Supp. Fig. 2B) does not involve any specific chromatin-chromatin or chromatin-envelope attractive forces. Hence, this self-organized state remains highly dynamic, and bears the hallmark features of a liquid crystal[45]. Such liquid crystal phases are commonly observed in dense solutions of semiflexible polymers[44,46], and typically arise from the local tendency of stiff polymer segments to spontaneously align along a common axis to optimize packing. In our case, the presence of a spherical envelope interferes with this effect, preventing the establishment of a unique, uniform direction of fiber alignment throughout the entire nuclear interior. Thus, our cylindrically-symmetric spool may be generically explained in terms of the interplay between spherical confinement and orientational, liquid-crystalline order[44].

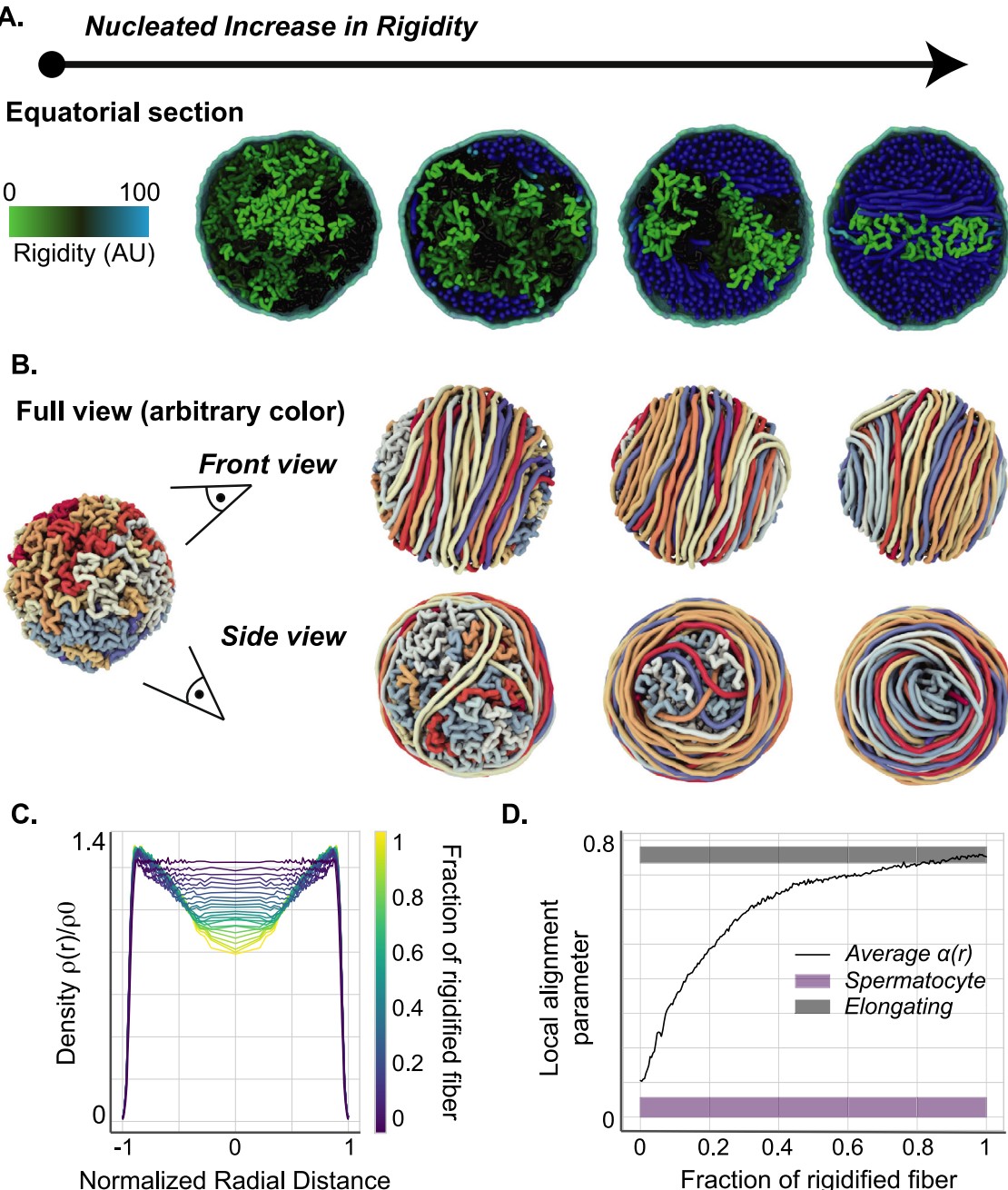

**Fig. 3 | Nucleated chromatin rigidification results in a spool-like organization.**
**A** Chromatin re-organization via a nucleated, *in cis* propagation of fiber rigidification. Fiber segments undergoing gradual stiffening are marked in green, denoting a putative peak in acetylation activity. Fully-rigidified segments are depicted in blue, representing SNBP-based chromatin. Unmodified chromatin regions are rendered in black. **B** Same as Fig. 3a in full-fiber view. The segregation of rigidified chromatin segments into a toroidal structure at the nuclear periphery is clearly visible, and leads to the formation of a spool-like ordered structure towards the late round spermatid stage. **C** Simulated radial monomer density profiles at different rigidified fiber fractions. Distance is normalized to nucleus radius. Nucleated rigidification results in a significantly more pronounced peripheral accumulation of chromatin than the simple uniform stiffening process (Fig. 2E), although the mechanical properties of the fully-rigidified fiber are identical in both cases. **D** Radially-averaged local alignment parameter $\langle \alpha(r) \rangle$, as computed from the simulations (black line) and evaluated from TEM images at the spermatocyte (purple box) and elongating stage (gray box). The width of the experimental bars represents the standard error of the mean (see "Methods" section).

## Chromatin orientation follows cell polarization

The nucleation model results in a spool that features a distinct axis imposed by the orientation of the spooled chromatin fiber. We thus wondered whether the definition of the antero-posterior axis of the cell preceded or rather was determined by chromatin orientation. Since the flagellum is a clear marker of cell posteriority, we performed immunofluorescent staining on α-Tubulin and observed that flagellar microtubules start forming in early round spermatids (Fig. 5A),

indicating that these cells already possess an antero-posterior axis at these stages. Further investigating electron micrographs of round spermatids, we identified cases where the nascent flagellum and acrosome were visible in multiple cells of the same cyst (Fig. 5B). At this stage chromatin displayed a circular symmetry, with the periphery being denser than the center portion of the nucleus (Fig. 5B). The lack of chromatin orientation in these polarized cells importantly shows that sperm cells acquire an antero-posterior axis before chromatin

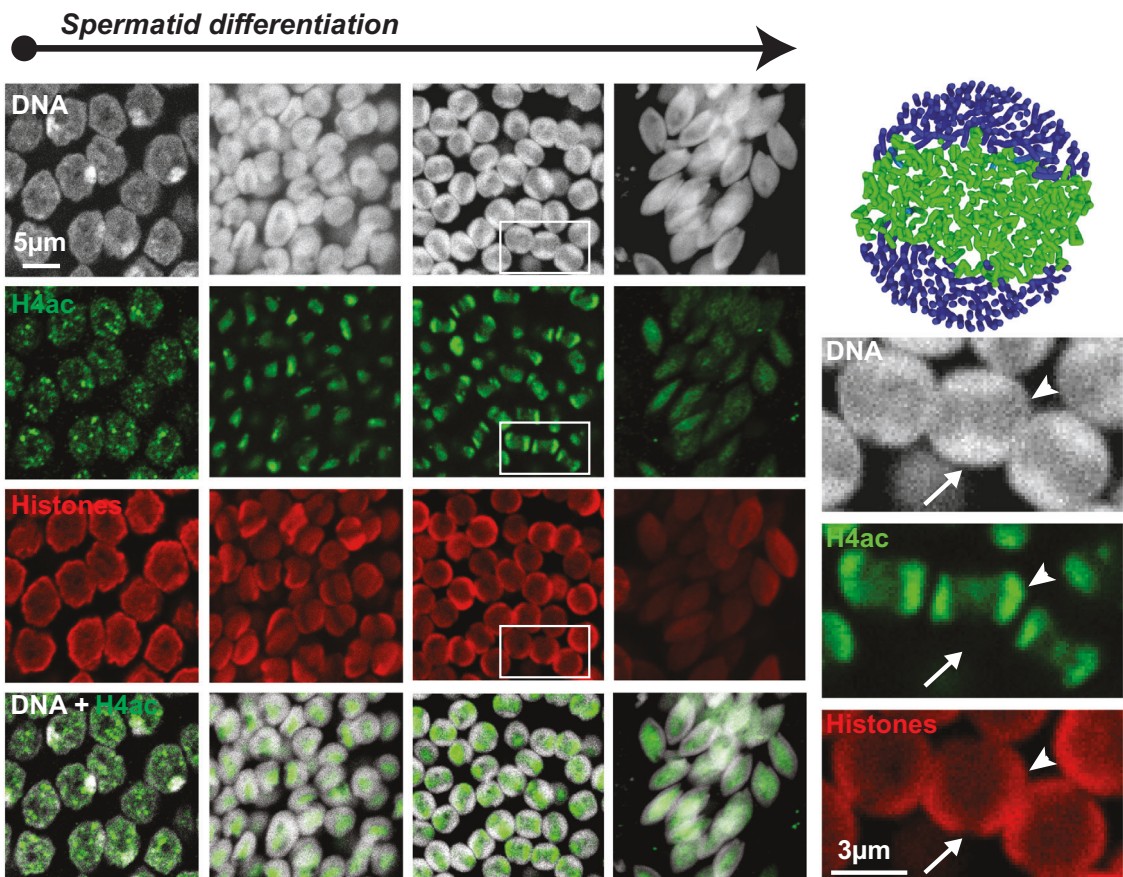

**Fig. 4 | Histone acetylation patterns in vivo are consistent with the rigidifica-tion dynamics predicted by our model.** Confocal images of spermatids at four early, consecutive stages in spermiogenesis stained for DNA (white), pan-acetylated histone H4 (H4ac, green), and total histones (red). H4ac appears as punctate nuclear foci that cluster at the nucleus center, subsequently spreading as two waves directed towards opposite nuclear poles (see arrowheads in magnified panels on the right). At this stage, DNA is enriched in the complementary nuclear space surrounding the central region (arrows). This organization is consistent with our simulations. Early elongating spermatids (right panels) show dispersed and weaker H4ac and histone signals.

does, suggesting that coiled fibers can reorganize to follow this axis in elongating nuclei.

To test whether chromatin orientation adapts to cell elongation, we similarly introduced an external stretching force acting on the spherical envelope encapsulating our spooled, stiff polymer (Figs. 2 and 5C). Our simulations show that the spooled fibers may readily align and stretch along the arbitrary elongation axis of the envelope, regardless of their own initial orientation (Fig. 5C, D). This alignment results from the expected tendency of rigid polymers to minimize bending by arranging along the directions of minimal curvatures. The nucleation-based model of chromatin rigidification and spooling thus better recapitulates the unidirectional peripheral alignment of the fiber that we observed in early canoe spermatids (Fig. 2D). We analyzed such large-scale chromatin rearrangements by computing the evolu-tion of the bulk alignment parameter $\alpha_{bulk}$, which quantifies the large-scale nematic order within the nuclear interior, as a function of the applied force $F$ (Fig. 5E, see Methods). We observed a transition from low $\alpha_{bulk}$ values ($\cong 0$) at weak $F$ (15 nN), indicating the lack of a pre-ferred global axis of alignment in the spooled configurations simulated in spherical confinement (Fig. 3A, B), to $\alpha_{bulk} \cong 0.75$ for $F \cong 25$ nN, compatible with that estimated from longitudinal electron micro-graphs of elongating spermatids (Fig. 5E and Supp. Fig. 3). Transversal sections of this elongating configuration nonetheless reveal that the polymer retains a distinctive organization reflecting that of the previous spool-like configuration (Figs. 3A and 5D). Indeed, chromatin is found to stretch along the nuclear antero-posterior axis by forming two distinct peripheral domains separated by a core region - corresponding to the vestigial spool axis - featuring an orthogonal direction of fiber align-ment. Supporting our simulations, we identified a strikingly similar configuration in electron microscopy images of elongating nuclei (Fig. 5D). These results further suggest that there is a temporal overlap between progressive chromatin stiffening and nuclear elongation, which is supported by the fact that H4ac and histones in general can still be detected in early elongating nuclei (Fig. 4). Together, we concluded that the orientation of coiled fibers follows the acquisition of the cell antero-posterior axis.

## A twisted fiber configuration is observed in mature sperm

Because of the extreme compaction in maturing sperm, electron microscopy only detects a homogeneous mass of dense material in their nuclei. To understand chromatin organization in the latest stages of spermiogenesis, we therefore turned to super-resolution fluorescence microscopy. We exploited a Hydroxymethyl-Silicon-Rhodamine (HMSiR) fluorophore coupled to the DNA dye Hoechst33382[47] that readily labeled DNA in situ in whole mount testes. We performed single-molecule localization microscopy (SMLM) detecting this dye and reconstructed high-resolution images of spermatid nuclei (Fig. 6A, B). At advanced stages in spermato-genesis, SMLM revealed an additional level of chromatin organiza-tion. Indeed, in late canoe cells, we observed that DNA was twisted around the antero-posterior axis within the nucleus (Fig. 6A). We further confirmed that this helical organization was also present in fully mature sperm cells collected from dissected female sper-matheca (Fig. 6B). Interestingly, previous work provided evidence for

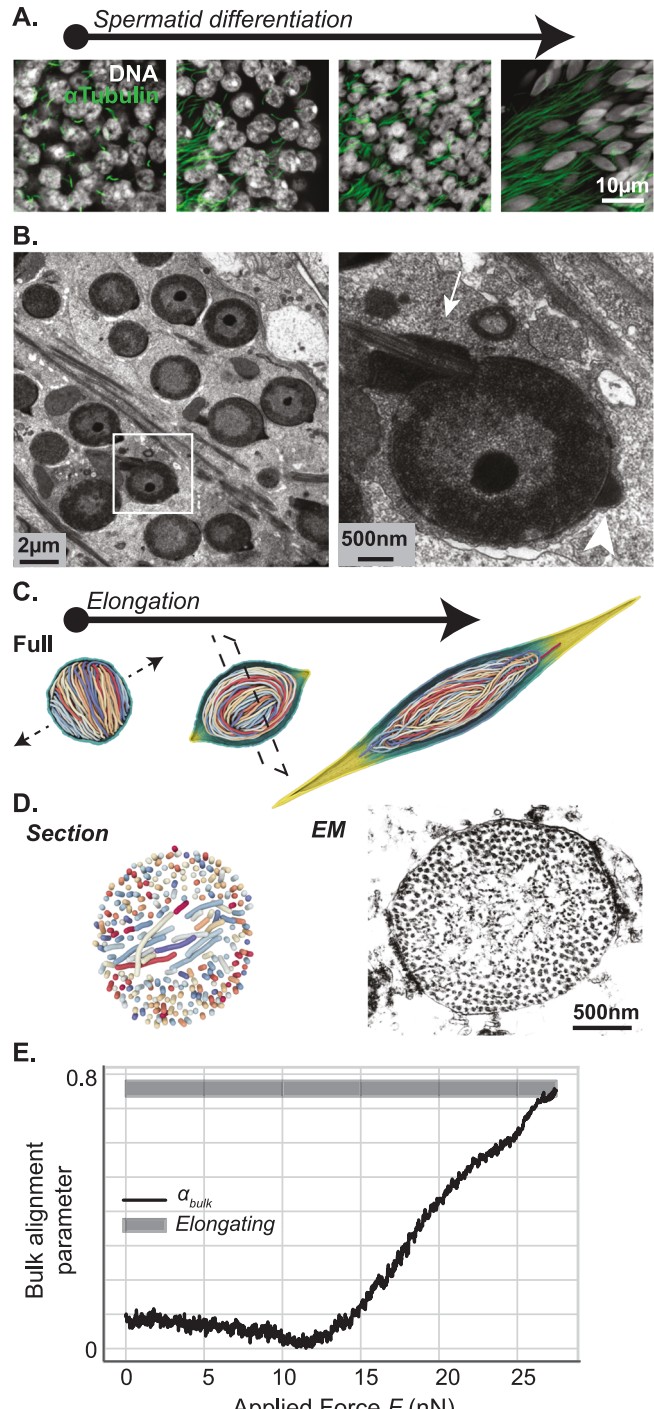

**Fig. 5 | Cellular polarization precedes chromatin orientation. A**, **B** Round spermatids feature nascent flagella and acrosomes. **A:** Confocal microscopy images of testes stained for DNA (white) and αTubulin (green) show nascent flagellar microtubules in early spermatids. **B** Electron microscopy image of a round spermatid cyst showing multiple nuclei (left). The boxed region is zoomed in the right panel. Chromatin is homogeneously enriched at the nuclear periphery, flanked by nascent flagellar (arrow) and acrosomal (arrowhead) structures, indicating that cellular antero-posterior poles were defined before chromatin became oriented. **C** In polymer simulations, nuclear elongation drives the spontaneous reorientation of the chromatin spool axis perpendicular to the antero-posterior line, and leads to the distinct unidirectional alignment of the chromatin fiber coils. **D** Elongating spool organization observed in simulations and in vivo. Transversal sections of nuclei during elongation in polymer simulations (left) and in electron microscopy images of elongating spermatids (right) show a similar configuration whereby the vestigial spool axis perpendicular to the elongation axis is still apparent. **E** Alignment parameter $\alpha$ computed through the bulk of the nuclear interior (see Methods), as a function of the applied stretching force $F$. As in Fig. 3D, bars correspond to the standard error of the mean from experimental measurements from EM images at the elongating stage (gray box).

(elongation and torsion), the twisted polymer spontaneously tightened very rapidly. This likely results from the polymer accommodating the mechanical constraints imposed initially by the torsional deformation, thus effectively capturing the impact of the actual forces driving chromatin twisting – although the latter were not explicitly modeled. This effect resulted in a highly compact structure consistent with that observed in vivo in mature sperm (Fig. 6C). This model suggests that the needle-like shape of the sperm nucleus could thus be further promoted by a lateral compaction effect induced by large-scale chromatin twisting. Searching for a putative mechanism for chromatin twisting, we confirmed previous observations that the basal part of the flagellum runs parallel to the nucleus[39]. At the flagellum-nucleus junction, we noticed a structure reminiscent of a cytoskeletal mesh in physical contact with chromatin (Fig. 6D). Furthermore, we observed that the basal-most end of the flagellum is completely embedded in nuclear material (Fig. 6D, see also Fig. 1D). Further supporting flagellar activity in elongating cells, we found that near-mature sperm occasionally displayed a coiled nuclear shape (Fig. 6E), which may result from a torsional force applied to compacted, needle-shaped chromatin. Hence, we speculate that unidirectional rotations of the flagellum might transmit a torque to chromatin via physical contacts and/or viscous drag effects, and thus establish its twisted configuration and further deform the nucleus as a whole when chromatin becomes highly packed. Together, these observations reveal an additional level of sperm chromatin organization whereby spooled fibers twist around the nuclear axis in late spermiogenesis, potentially via a torque imposed by the flagellum, which may ultimately facilitate compaction.

## Disulfide bonds stabilize the ultracompact chromatin organization

In the simulations described above, release of the applied extensional force and geometrical constraints resulted foremost in polymer ultracompaction, but this configuration was in fact unstable and transient. Indeed, following this initial step, the nucleus readily transitioned back into a relaxed state (Fig. 7A). We took advantage of this situation to make a parallel with the chromatin decondensation dynamics that occur at fertilization, which can also be achieved experimentally by chemically treating spermatozoa with reducing agents. In particular, exposing *Drosophila* mature sperm to Dithiothreitol (DTT) causes reduction of disulfide bonds between Protamines, their eviction from chromatin, and a rapid decompaction of the needle-shaped nucleus[29]. In our simulations, the unconstrained needle-shaped polymer as a whole shortened through lateral swelling and curling (Fig. 7A). To test these predictions, we treated mature

a helical configuration of DNA in mature sperm of the cricket *C. nigricans*, suggesting a conserved feature[37]. Since early elongating nuclei did not display such twists, we inferred that the events underlying this higher-order structure occurred at the later stages of elongation.

Within our modeling hypotheses, we could not identify any simple and plausible changes in the polymer properties that could intrinsically generate this twist. Instead, we putatively implemented a geometric torsional field that, added to the elongation force, resulted in a twisted configuration (Fig. 6C). Importantly, a transversal section of this twisted polymer displayed a characteristic screw-like organization that was highly reminiscent of chromatin fiber organization in late spermatids visualized by electron microscopy (Fig. 6C). Interestingly, upon release of the imposed shaping forces in these simulations

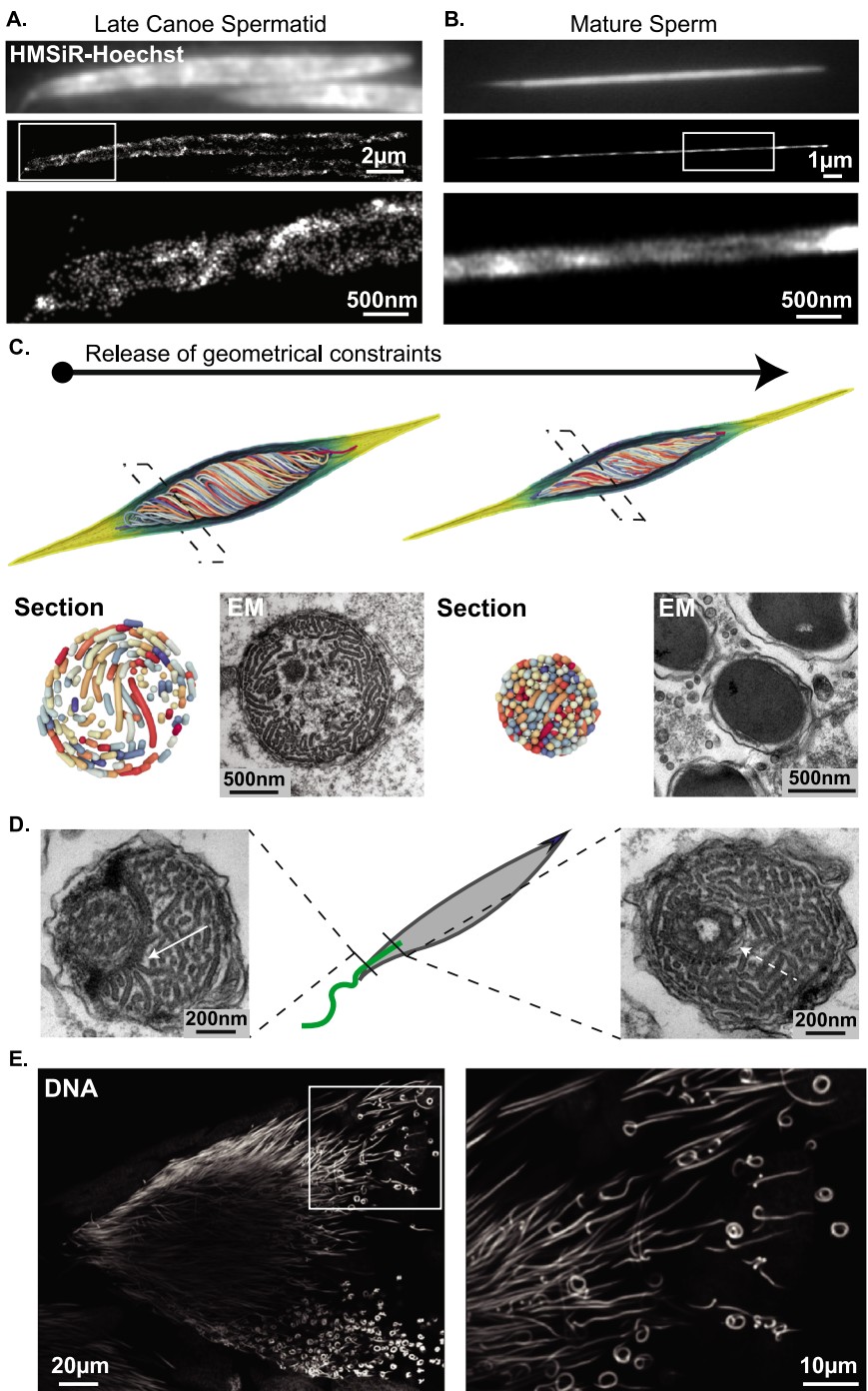

**Fig. 6 | Chromatin twists around the nucleus axis in late spermiogenesis.**
**A**, **B** DNA twisting in late spermatids. Single Molecule Localization Microscopy (SMLM) images of late canoe (**A**) and fully mature (**B**) spermatids labeled with an HMSiR-Hoechst probe to reveal DNA with nanoscopic resolution. Top panels show the epifluorescence image; middle and bottom panels show SMLM images and close in. DNA twists around the nucleus axis over several turns. Boxed regions are zoomed in bottom panels. **C** In simulated data, a geometric twist imposes an additional extensional strain on the peripheral chromatin fibers, which leads to an increased degree of lateral compaction upon mechanical relaxation. Chromatin configuration and density in simulated polymers before and after relaxation recapitulate that observed in vivo by electron microscopy in late canoe (bottom left) and mature (bottom right) spermatids respectively. **D** Evidence for contacts between the flagellum and chromatin. Electron microscopy transversal sections of the basal end of canoe spermatids reveal a proteinaceous structure connecting the flagellar axoneme and chromatin fibers (full arrow). Close to its organizing center, the flagellum fully penetrates the nucleus and is fully embedded by chromatin (dashed arrow). **E** Whole-nucleus looping in late spermatids. Confocal microscopy images of late needle-shaped spermatid cells in testes labeled for DNA with DAPI, showing occasional curling. The boxed region is zoomed in the right panel.

sperm isolated from spermatheca with 25 mM DTT for up to 30 min (Fig. 7B). Upon treatment, needle-shaped nuclei became progressively unwound and decompacted. Intermediate steps of chemically induced decompaction showed remarkable similarity with simulated decondensation steps and were fully compatible with the progressive loosening of the higher-order twisted organization. We conclude that the ultracompact needle configuration is locked in vivo in the absence of any external force, thus ensuring that sperm shape is preserved even when submitted to constraints such as flagellar movement, until the time of fertilization.

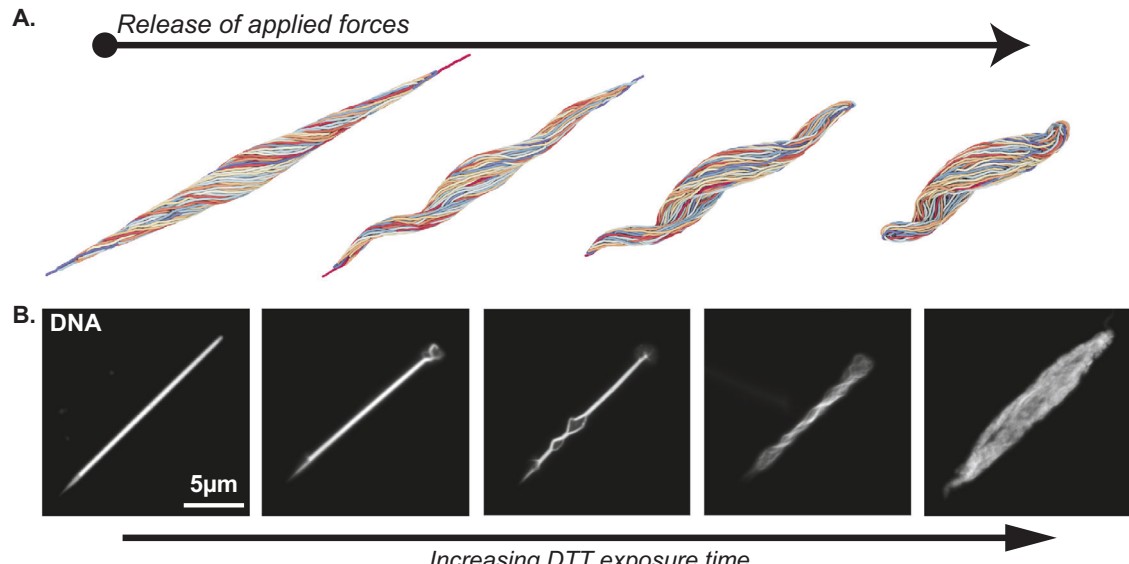

**A.** Release of applied forces

**B.** DNA

5μm

Increasing DTT exposure time

**Fig. 7 | Disulfide bridging maintains nuclear compaction and shaping. A** Nuclear decondensation is modeled by the release of applied extensional forces, and leads to a rapid relaxation towards a spherical envelope shape via a transient, corkscrew-like intermediary morphology. **B** Chromatin is readily decondensed upon treatment with reducing agent DTT. Epifluorescence images of mature spermatids isolated from female spermatheca, labeled for DNA and exposed to 25 mM DTT. Chromatin decondensation pattern is consistent with simulated data and suggests that protamine disulfide bridges maintain the needle-like nuclear morphology.

## Discussion

### *G. bimaculatus* offers a uniquely elegant solution to sperm genome packaging

Packaging the genome in a compact configuration during spermiogenesis is a universal problem for which the coil organization in *G. bimaculatus* is a simple and elegant solution. Suzuki & Wakabayashi[39] first revealed the existence of a unique genome arrangement as bundled, thick chromatin fibers in canoe stage *G. bimaculatus* sperm. In the present study, we validate all the observations made in this pioneering work, and further provide a complete sequential view on how chromatin organization is rearranged from spermatocytes to mature sperm, as well as mechanistic insight into how these dynamic transitions operate. The resulting twisted spooled coil arrangement maximizes space occupancy while minimizing chromosome entanglement, thus ensuring ultracompaction while facilitating genome unfolding at fertilization.

Our simulations reveal that this ordered configuration can be achieved through relatively simple changes in the biophysical properties of chromatin, such as a nucleated rigidification of the fiber. In mice and *Drosophila*, a nucleus-wide wave of histone acetylation precedes their replacement with SNBPs, which may be responsible for chromatin stiffening[14,15,20]. The ~25–30 nm thick chromatin observed in sperm cells of multiple insects, including that of *G. bimaculatus*, fed a model for genome folding whereby a "30 nm fiber" was considered as a universal level of chromatin organization[48]. While this model is now outdated, it is noteworthy that such fibers are frequently observed in insect sperm but were also found in the sperm of other invertebrates, including for example sea urchins, which do not bear protamines but rather incorporate sperm-specific histone variants on maturing sperm, or cephalopods[49,50]. While theoretical work has proposed models on how DNA charge neutralization may favor this configuration, how such rigid fibers are formed in vivo remains largely unknown. Our study only indirectly addresses the existence and function of TPLs and PLs in *G. bimaculatus*: their identification would be required to further address how they may organize the ~25 nm sperm chromatin fiber. Yet, former studies suggest that there could be only a single SNBP in mature cricket sperm[39] - in contrast to two in most mammals and over five in flies - making this a simplified model potentially more amenable to investigate these issues.

Illustrating the importance of these mechanisms, failure in histone hyper-acetylation correlates with infertility in mice and men[51,52] and blocks histone replacement and sperm differentiation in *Drosophila*[53]. In mice, histone hyperacetylation is thought to result from the activity of HATs including NUT-dependent p300 and CBP as well as Gcn5[23,54], favored by a decrease in the levels of histone deacetylases (HDACs) in sperm[18,55]. Accordingly, NUT or Gcn5 mutants display partial failure in histone-protamine replacement, together with aberrant sperm compaction and shaping. In crickets, we further show a histone acetylation pattern consistent with nucleation and spreading of this modification. The spreading of histone modifications has been well described for repressive heterochromatin methylation marks such as lysine 9 and lysine 27 on histone H3. In mice and *Drosophila*, enzymatic complexes can both establish and recognize the methylation marks, therefore feeding a spreading mechanism on neighboring nucleosomes. These modifications alter the mechanical properties of chromatin and attract additional complexes, which favors the formation of nuclear sub-compartments[56–60] where these processes are further catalyzed[61]. Histone acetylation can analogously target DNA regulatory elements (notably enhancers and promoters) via DNA sequence-specific transcription factors recruiting HATs and bromodomain proteins[62,63]. It was further proposed that a feedforward loop involving NUT, p300, and BRDT helps spreading histone acetylation in mouse sperm[23]. The targeting mechanisms for HATs acting on mouse or *Drosophila* sperm is unknown but the rather homogeneous nuclear distribution of acetylation favors a model in which histones are randomly and synchronously acetylated as a nucleus-wide wave, with the notable exception of genomic regions resisting the histone-protamine transition[20]. Yet, these mechanisms must differ in cricket sperm, where early acetylation displays a sharp spatially defined sub-nuclear pattern. The molecular bases for such nucleation of acetylation are unknown but may be key to account for the strikingly different genome reorganization in crickets compared to mice and *Drosophila*. Identification of the enzymes responsible for acetylation in crickets, as well as the genomic distribution of histone acetylation and HAT binding during spermiogenesis is needed to further explore these questions.

The relative simplicity of the biophysical forces we uncover makes it all the more intriguing that, to our knowledge, this specific chromatin configuration has not been reported in any other cell type or

species thus far. We propose that the precise spatio-temporal patterning of acetylation and histone-SNBP replacement in sperm cells is critical to determine chromatin organization. Notably, no particular higher-order genome organization is observed in electron micrographs from *Drosophila* sperm[64], which features a homogeneous histone hyperacetylation distribution. In mice, Hi-C maps of DNA-DNA contacts have yielded contrasting results, but usually revealed relatively modest differences in sperm cells compared to somatic cells[65–67]. These surprising observations imply that the massive histone-SNBP replacement does not cause large-scale genome 3D rearrangements in this species. While these studies need further confirmation, developments of our modeling framework could help identify how histone acetylation and replacement patterns may result in this intriguing preservation of genome organization, which must clearly not occur in crickets.

At later stages of spermiogenesis, the twisting of the chromatin fiber spools may contribute to achieve maximum compaction. While we could not unequivocally identify its mechanistic bases, we suggest that unidirectional rotational movements of the flagellum may be involved. A limited number of turns may be enough to transmit sufficient rotational forces onto chromatin and the nucleus as a whole to explain the observed deformations. This finding would imply a nuclear structuring role for the flagellum, in addition to its motility and sensory roles. Whether flagellar components interact with chromatin and how they could exert such mechanical forces would deserve further experimental interrogation. Importantly, our results also suggest that additional processes may be required towards the final steps of spermiogenesis in order for sperm cells to acquire a robust needle-like morphology despite this persisting torque. While the cricket SNBPs remain to be identified, we speculate that the effect of DTT in mature sperm may be due to reduction of disulfide bonding among these proteins. Future work should thus aim at identifying whether cricket SNBPs bear cysteine residues and whether their intermolecular bonds might serve to stabilize such a nuclear configuration, so that flagellar movement may drive cellular motion without altering cell shape.

### Sperm chromatin rearrangements are consistent with liquid-crystalline behavior

The spool-like chromatin arrangement obtained at the outset of the rigidification of the full fiber is driven by the spontaneous alignment of neighboring chain segments within the near-spherical nuclear envelope, and arises from the sole competition between excluded volume and filament bending energy. It may therefore be identified as a self-organized, liquid-crystalline (nematic) state, characterized by the conjunction of local orientational order with a dynamic, liquid-like mode of molecular motion[45]. Previous theoretical works have already suggested that the coupling between polymer stiffness and confinement may lead to liquid-crystal-like folding patterns[43,44,68–71]. For example, a configuration known as the condensed Hopf fibration, similar to our spooled chromatin arrangement (Fig. 3B), has been recently predicted by a mean-field theory for long, semi-flexible polymers in tight spherical confinement[68]. Our work suggests that the rigidification of chromatin fiber via the nucleated and progressive increase in polymer stiffness, as motivated by our original biological observations, provides a particularly optimal kinetic pathway to achieve such ordered ground states starting from a random, dense polymer organization in confinement.

The unique combination of order and fluidity provided by this original packing mechanism may in our case play a central functional role in the maturation of cricket sperm cells. Indeed, our results suggest that the strong orientational alignment of the chromatin fiber, associated with its ability to dynamically reorganize following changes in nuclear morphology and the application of torsional forces, may be key to achieve maximal levels of compaction and elongation in mature sperm nuclei. Hence, our findings would support a model in which

irreversible intermolecular interactions, potentially induced by disulfide protamine bonds, should be established at the very final stage of spermiogenesis, in order to "freeze" the resulting liquid-crystalline chromatin assembly into a crosslinked state resilient to external perturbations.

In this context, the potential influence of the detailed fiber structure on the observed chromatin arrangements would warrant further investigations. For instance, the chiral symmetry of many liquid-crystal-forming biopolymers such as DNA is generally known to promote twisted, non-uniform patterns of orientational order, which may in turn give rise to a higher level of hierarchical organization[72]. These so-called cholesteric liquid crystal phases are commonly observed in the mature sperm heads of many vertebrates[73], and have been more broadly proposed to fulfill a variety of biological purposes and functions[74]. Such cholesteric interactions could in our case serve to further stabilize the twisted fiber configuration uncovered in mature spermatids, and thus contribute to the establishment and maintenance of optimal packing and elongation in the cricket sperm. Interestingly, similar cholesteric arrangements have also been reported in dinoflagellate chromosomes[75], which are characterized by unusually-large genome sizes, and feature a near-total absence of histones and histone-like proteins[76]. Therefore, our results would more generally suggest the combination of histone replacement and liquid-crystalline organization as a convergent evolutionary process for the ultradense packaging of chromatin within highly-confined nuclear environments.

## Methods

### Cricket breeding and sample collection

The *Gryllus bimaculatus* white-eyed mutant strain was a gift from C. Extavour and T. Mito. Crickets were reared at constant 29 °C under a 12-h light–dark photoperiod, fed ad-libitum with powdered mouse food, quenched with tap water, and allowed to lay eggs in soaked tissue paper. For imaging experiments and protein extraction, pairs of testes were dissected from sexually active adult males, while mature sperm was isolated from adult female spermatheca. Animals were anesthetized with CO2 prior to dissection.

### Confocal microscopy

For immunofluorescence staining, testes were dissected in PBS 0.1%, Triton X-100 (PBS-T) and fixed at room temperature in 4% formaldehyde in PBS for 30 minutes. Testes were washed three times (10 min each) with PBS-T and incubated with anti-H4ac (Merck Millipore #06-598; 1/500 dilution), anti-Histones (Millipore #MABE71; 1/2500 dilution) or anti-αTubulin (Sigma #T9026; 1/500 dilution) primary antibodies in the same buffer on a wheel overnight at 4 °C. They were then washed three times (10 min each) with PBS-T and secondary antibody incubations using DyLight 550 conjugated goat anti-mouse IgG (ThermoScientific #84540; 1/1000 dilution) and DyLight 488 conjugated goat anti-rabbit IgG (ThermoScientific #35552; 1/1000 dilution) were performed identically. Testes and sperm samples were mounted in DAKO mounting medium containing DAPI (2 µg/ml). Images were acquired on an LSM 800 confocal microscope (Carl Zeiss). Images were processed with Zen imaging software (Carl Zeiss) and FIJI software[77].

### Single molecule localization microscopy

Fixed testes or sperm samples were mounted in PBS medium containing 100 nM Hydroxymethyl Silicon-Rhodamine-Hoechst probe We acquired the streams using MetaMorph software (Molecular devices) on a home-built setup consisting in an IX81-ZDC2 microscope equipped with a temperature-controlled incubation cage kept at 27 °C. The intensity and wavelengths of the illumination were controlled through Labview to pilot an acousto-optical tunable filter (AOTF; Quanta Tech). Wide-field illumination was achieved by focusing the laser beams to

the back focal plane of a 100x 1.49 numerical aperture (NA) oil-immersion apochromatic objective lens (Olympus). Data collection was obtained using an Evolve EMCCD camera (Photometrics) with continuous 642 nm light illumination, at high power. Images were reconstructed using the ThuderSTORM[78] Fiji plugin.

## Transmission electron microscopy
Testes were dissected in water and fixed for 20 h at 4 °C in 2% glutar-aldehyde and 2% paraformaldehyde in 0.1 M sodium cacodylate pH 7.35. After four washings of 12 h at 4 °C in 0.15 M sodium cacodylate, samples were postfixed 1 h at RT in 1% OsO4 / 0.15 M sodium cacody-late. They were dehydrated in ethanol solutions (30°/50°/70°/80°/95°/100°) for 30 min each, and in propylene oxide 2 times 15 min. After 6 baths of substitution and 3 baths of impregnation, samples were embedded in epoxy resin in flat silicon molds and polymerized at 56 °C for 48 h. Ultrathin sections were cut with a UC7 Leica ultramicrotome. Ultrathin sections were contrasted in uranyl-acetate and lead citrate solutions. Sections were observed with a Philips CM120 transmission electron microscope at 120Kv (Centre Technologique des Micro-structures, Université Lyon 1, Lyon, France).

## Western blotting
Testes or sperm samples were homogenized by roto-douncing in 5% SDS, 50 mM TEAB, 50 mM DTT. The homogenates were incubated overnight at 37 °C, pelleted and clear lysates were collected and quantified. 1 or 2 μg of total protein were separated in 4–12% acryla-mide gels, transferred to PVDF membranes and stained with Pierce Reversible Protein Stain Kit for total protein visualization. Stain was washed away, membranes were blocked in 5% non-fat milk and stained with an anti-H3 antibody (abcam #1791, 1/2500 dilution, overnight incubation at 4 °C) in TBS-T with 1% non-fat milk. Membranes were washed in TBS-T, stained with HRP-coupled secondary antibody (Agilent Dako #P044801-2; 1/10000 dilution) for 2 h at room temperature, and washed again with TBS-T. Proteins were revealed by chemilumi-nescence using Pierce SuperSignal West Pico substrate and imaged in a ChemiDoc imaging system.

## Chemically induced decondensation assays
Sperm samples were isolated from female spermatheca, spread in PBS onto microscopy slides. Excess liquid was removed and sperm was mounted in PBS-Triton 0.3% containing 5 μg/ml of Hoechst with 25 mM DTT for immediate visualization.

## Statistics and reproducibility
Each electron micrographs shown is a representative example from at least 10 similar observations from 3 independent experiments. Con-focal microscopy images were chosen as representative from at least 10 similar observations in at least 2 independent experiments. STORM images shown are representative of an ensemble of >20 images collected from 4 independent experiments. Western blots were performed 3 independent times with similar results.

## Numerical model
The nuclear envelope was described as a cross-linked biopolymeric shell consisting of a fixed number $N_v$ of vertices, which were initially distributed uniformly on the surface of a sphere of radius $R$[44]. Pairs of neighboring vertices were linked via springs of stiffness $k_{memb}$ and rest length $r_0$, as represented by the harmonic potential

$$U_{memb}^{bond}(r_{vv}) = \frac{k_{memb}}{2}(r_{vv} - r_0)^2,$$

with $r_{vv}$ the inter-vertex distance. The set of connected vertices, which defines the topology of the envelope, was constructed by Delaunay triangulation of the initial vertex positions through the Quickhull

algorithm[79], and was kept fixed throughout the simulations to mimic the static cross-links of the nuclear envelope lamina network[80]. Self-avoidance was achieved through the use of a truncated and shifted Lennard-Jones potential with exclusion radius $r_0$ and depth $\varepsilon$,

$$U_{(r_0)}^{exc}(r)\begin{cases} 4\varepsilon\left[\left(\frac{r_0}{r}\right)^{12} - \left(\frac{r_0}{r}\right)^6 + \frac{1}{4}\right], & r < 2^{1/6}r_0 \\ 0, & r \geq 2^{1/6}r_0 \end{cases}, \quad (1)$$

which was applied to all pairs of unconnected vertices.

The chromatin fiber was modeled as a single linear polymer comprised of $N_m$ monomeric subunits of effective diameter $\sigma$, joined together by harmonic springs of stiffness $k_{chrom}$,

$$U_{chrom}^{bond}(r_{mm}) = \frac{k_{chrom}}{2}(r_{mm} - \sigma)^2,$$

with $r_{mm}$ the distance separating a pair of consecutive monomers along the chain. Excluded volume effects were emulated through the use of a soft repulsive potential of depth $\varepsilon_{chrom}$,

$$U_{chrom}^{exc}(r) = \begin{cases} \varepsilon_{chrom}\left[1 + \left(\frac{r}{\sigma}\right)^{12}\left(\frac{6r^2}{\sigma^2} - 7\right)\right], & r < \sigma \\ 0, & r \geq \sigma \end{cases},$$

which similarly acts on all pairs of non-contiguous monomers. The bending rigidity of the chromatin fiber was represented by a standard angular potential of stiffness $k_{bend}$,

$$U_{chrom}^{angle}(\theta) = k_{bend}(1 - \cos\theta), \quad (2)$$

where $\theta$ denotes the angle formed by the two bonds linking a triplet of adjacent monomers. The mutual repulsion between chromatin and the encapsulating envelope was finally described by an interaction term $U_{\Sigma}^{exc}$ of the same form as Eq. (1) for each monomer-vertex pair. In this case, the exclusion radius was set to $\Sigma = r_0 + \sigma$, which effectively ensured that the envelope surface could not be crossed by the chromatin chain in any configuration. For simplicity, we neglected the explicit role of nucleoplasmic and hydrodynamic forces on monomer dynamics[81].

The effective diameter of the chromatin fiber was set to $\sigma = 30\,nm$, which serves as the model unit of length. Each monomer thus contains about ~15 kbp, based on our estimation of the fiber linear DNA density (see main text). For computational feasibility, a number $N_m = 10,000$ of monomeric units were considered, which corresponds to a chromatin fiber length $L = N_m\sigma = 300\,\mu m$. This value amounts to a fraction $f \sim 10\%$ of the total chromatin content, as estimated by electron microscopy imaging of canoe-stage spermatids (see main text). Accordingly, the initial envelope radius was set to $R = R_{round} \times f^{1/3} = 0.7\,\mu m$ to ensure the conservation of chromatin density, where $R_{round} \cong 1.5\,\mu m$ is the experimentally-measured nuclear radius at the early round spermatid stage. An arbitrarily-high number $N_v = 5,560$ of vertices were used in the calculations to provide a fine-mesh discretization of the envelope surface, and the spring rest length was set to the associated mean inter-vertex distance in the uniform initial state, i.e., $r_0 = (8\pi R^2/\sqrt{3}N_v)^{0.5} \cong 35\,nm$.

Molecular dynamics (MD) simulations were run on multiple graphics processing units via the HOOMD-blue software package[82,83]. A standard Langevin thermostat was used to thermalize the system at $T = 293K$, in order to both emulate the effects of thermal fluctuations as well as to effectively account for other putative random forces emanating from active processes involving chromatin[84]. The model unit of energy was set to $\varepsilon = k_B T$, with $k_B$ the Boltzmann constant. Correspondingly, a height $\varepsilon_{chrom} = 5\varepsilon$ was employed for the chromatin-chromatin repulsive barrier, and the stiffness of the chromatin back-bone springs was taken as $k_{chrom} = 15\varepsilon/\sigma^2$. This choice of parameters was specified to allow for occasional self-crossings of the chromatin

chain, and thus mimics moderate topoisomerase activity. The stiffness of the envelope springs was set to $k_{memb} = 2500\varepsilon/\sigma^2 \cong 10\,mN/m$, which matches the typical Young's modulus of the bare nuclear lamina network estimated from micropipette aspiration experiments[85]. A vanishing bending rigidity $k_{bend} = 0$ was assumed in the initial state to emulate the high compliance of the nucleosomal chromatin fiber at the onset of the round spermatid stage.

## Relaxation and rigidification runs

The chromatin chain was initialized as a dense, self-avoiding random walk encapsulated within the bounds of the initial spherical envelope conformation. Relaxation runs of $\mathcal{O}(10^8)$ MD steps were performed to achieve a homogeneous chromatin density throughout the nuclear interior. For simulations of uniform fiber rigidification, the chromatin bending modulus was gradually raised from $k_{bend} = 0$ to $k_{bend} = 50\varepsilon$ in $\mathcal{O}(10^2)$ linear increments. This latter value corresponds to a chromatin bare persistence length $l_p \cong 1.5\,\mu m$, qualitatively consistent with electron microscopy observations of the fiber at the canoe stage (Fig. 2D). $\mathcal{O}(10^8)$ MD steps were performed following each parameter update in order to reach a stationary state.

In the case of nucleated stiffening simulations, all loci located within a distance $d = 150\,nm$ of the center of the nucleus – corresponding to about 2% of the total chromatin content – were tagged in the relaxed initial configuration to mimic the transient action of a putative localized acetylation factory. A state-dependent bending rigidity $k_{bend}$ was then used, in which the stiffness of all bonds involving tagged loci was similarly raised from 0 to $50\varepsilon$ in an incremental fashion over $\mathcal{O}(10^8)$ MD steps, while the bending modulus associated with triplets of adjacent untagged loci remained set to $k_{bend} = 0$ (c.f. Eq. (2)). This process emulates a progressive rigidification of the chromatin fiber through the gradual eviction of acetylated histones, as discussed in the main text. Upon full stiffening of the tagged genomic regions, a number $N_{spread} = \mathcal{O}(10)$ of consecutive monomers were subsequently tagged on either side of the boundaries of each acetylated domain, to reproduce a slow *cis*-spreading of acetylation marks along the chain. The above protocol was then repeated over $\mathcal{O}(10^2)$ iterations, until the complete rigidification of the whole fiber was achieved.

## Nuclear elongation simulations

To simulate the directional elongation of the nuclear envelope, two antipodal vertices were identified along an arbitrary stretching axis $\vec{u}_{stretch}$ in the final configuration of the chromatin rigidification runs. An outward-directed force gradient $\vec{F}_{stretch} = \pm ||\vec{F}_{stretch}|| \times \vec{u}_{stretch}$ was then applied in the vicinity of the two vertices, whose magnitude was given by $||\vec{F}_{stretch}|| = F$ at each extremal vertex, and $||\vec{F}_{stretch}|| = F/2$ at each of their respective connected neighbors. The magnitude of the force was raised from $F = 0$ to $F = 2 \times 10^5 \varepsilon/\sigma \cong 27\,nN$, consistent with typical values reported in nuclear stretching experiments[85], over $\mathcal{O}(10^2)$ iterations. The system was similarly evolved over $\mathcal{O}(10^8)$ MD steps following each force increment.

## Twisting and decondensation simulations

To model the effects of geometric twist on spermatid morphology, a uniform torsional field of axis $\vec{u}_{stretch}$ and total twist angle $10\pi$ was applied to the simulated nuclei at various levels of elongation, which yielded a local thread angle compatible with single-molecule localization microscopy observations of late-stage spermatids (Fig. 5). The system was then allowed to relax over $\mathcal{O}(10^5)$ MD steps, which resulted in a significant lateral compaction of the encapsulated chromatin (Fig. 5C). For decondensation simulations, the magnitude $F$ of the external force acting on the envelope vertices was subsequently set to 0, and the kinetics of nuclear recovery towards the spherical reference elastic state of the envelope were monitored over $\mathcal{O}(10^7)$ MD steps.

## Quantification of orientational orders

The degree of alignment was obtained from our simulations via the eigenvalues of the nematic (Landau-de Gennes) Q-tensor,

$$Q_{\alpha\beta} = \frac{3t_\alpha t_\beta - \delta_{\alpha\beta}}{2}, \tag{3}$$

where $t_{\alpha,\beta}$ denotes the components of an arbitrary normalized bond vector $\boldsymbol{t}$ and $\delta$ is the Kronecker delta. For the estimation of position-dependent order parameters, the nucleus was partitioned into a number $\mathcal{O}(100)$ of spherical elements of identical volume, and Eq. (3) was separately averaged over all bonds $\boldsymbol{t}$ lying within each volume elements located at radial distance r from the nuclear center. The local alignment parameter was then defined as[44]

$$\alpha(\boldsymbol{r}) = \frac{2[\lambda_3(\boldsymbol{r}) - \lambda_2(\boldsymbol{r})]}{3}, \tag{4}$$

where the $\lambda_i(\boldsymbol{r})$ denote the ascending eigenvalues of the Q-tensor associated with an individual volume element centered at position $\boldsymbol{r}$ (Eq. (3)). The radial order parameter $\alpha(r)$ was then evaluated by averaging Eq. (4) over a spherical shell of radius $r$. Bulk (resp. surface) nematic order parameters were computed by averaging the Q-tensor (Eq. (3)) over all bonds located at radial distance $r < 0.85R$ (resp. $r > 0.85R$) and subsequently evaluating the corresponding eigenvalues. For the quantification of nematic alignment from electron microscopy imaging, statistical ensembles of $\mathcal{O}(10)$ TEM micrographs of spermatocytes and longitudinal sections of elongating spermatids (Supp. Figure 1A, D) were employed, in which the angular fiber distribution $P(\theta)$ was estimated using the FIJI OrientationJ plugin, with σ = 4 and Cubic Spline gradient parameters[86]. In this case, the nematic order parameter was defined as the largest eigenvalue of the averaged, two-dimensional nematic Q-tensor,

$$\left\langle Q_{\alpha\beta}^{2D} \right\rangle = \int_{-\pi}^{\pi} d\theta\, P(\theta)[2t_\alpha(\theta)t_\beta(\theta) - \delta_{\alpha\beta}], \tag{5}$$

with $\boldsymbol{t}(\theta) = \begin{pmatrix} \cos\theta \\ \sin\theta \end{pmatrix}$, which similarly describes the degree of orientational order projected within the 2D microscopic cross-sections. The integral in Eq. (5) was computed through standard quadrature methods, using a number $\mathcal{O}(100)$ of angular bins.

## Reporting summary

Further information on research design is available in the Nature Portfolio Reporting Summary linked to this article.

## Data availability

The imaging data generated in this study and used for analysis have been deposited in the Zenodo database under accession code https://doi.org/10.5281/zenodo.8043115.

## Code availability

Code used in this manuscript is available at https://github.com/mtortora/HOOMD-Nucl.git.

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

## Acknowledgements

We are grateful to Antoine Coulon, Raphaëlle Dubruille, Mylène Pezet, and Saadi Khochbin for fruitful discussions and critical reading of the manuscript. We thank Cassandra Extavour and Taro Mito for providing us with the white-eyed cricket strain. Work in D.J. group is funded by the Agence Nationale de la Recherche (ANR-18-CE12-0006-03, ANR-18-CE45-0022-01, ANR-21-CE13-0037-02). Work in B.L. group is funded by the Agence Nationale de la Recherche (ANR-21-CE13-0037-01 "Orthosperm") and by CNRS "Pack Modélisation" and "Pack Biodiversité". Work in G.A.O. group is supported by Agence Nationale de la Recherche (ANR-21-CE12-0014) and INSERM – Plan Cancer PCSI (ASC20020CSA). We thank the SFR Biosciences (UAR3444/CNRS, US8/Inserm, ENS de Lyon, UCBL) for imaging at the PLATIM, the CTµ microscopy facility of the Université Claude Bernard, the PSMN (Pôle Scientifique de Modélisation Numérique) and Centre Blaise Pascal of the ENS de Lyon for computing resources. We thank, Françoise Lacroix and Oleksandr Glushonkov (Institut de Biologie Structurale, Grenoble) for the support and access to the M4D Cell imaging Platform and thank FRISBI et GRAL for their UAR support. This work used the platforms of the Grenoble Instruct-ERIC center (ISBG; UAR 3518 CNRS-CEA-UGA-EMBL) within the Grenoble Partnership for Structural Biology (PSB), supported by FRISBI (ANR-10-INBS-0005-02) and GRAL, financed within the University Grenoble Alpes graduate school (Ecoles Universitaires de Recherche) CBH-EUR-GS (ANR-17-EURE-0003). J.B. and G.L. are funded by the Max Planck Society.

## Author contributions

G.A.O., D.J., and B.L. acquired funding, coordinated and supervised the project; G.A.O., M.M.C.T., B.H., D.J., and B.L. designed the research. G.A.O., M.M.C.T., B.H., D.B., J.P.K., D.J., and B.L. performed research. J.B. and G.L. contributed new reagents. G.A.O., M.M.C.T., B.H., D.J., and B.L. analyzed data. G.A.O., M.M.C.T., D.J., and B.L. wrote the paper with contributions from all authors.

## Competing interests

The authors declare no competing interests.
