## [Peer Review File · Nature Communications]

Biophysical ordering transitions underlie genome 3D re-organization during cricket spermiogenesis.REVIEWER COMMENTS

Reviewer #1 (Remarks to the Author):

This is a very interesting paper about the transitions undergone by chromatin during cricket spermiogenesis. In it, a biophysical model simulation based on polymer behavior is developed to account for the chromatin organization transitions observed in combination with electron, confocal and single molecule localization microscopy. The paper can certainly be considered for publication after some extensive rewriting.

COMMENTS: Chromatin modelling using polymer thermodynamic approaches had been previously used to account for the 25-30 nm diameter common to nucleoprotamine and nucleohistone chromatin organization regardless of their non-nucleosome or nucleosome internal organization. [Order and disorder in 30 nm chromatin fibers. Subirana JA. FEBS Lett. 1992 May 11;302(2):105-7. doi: 10.1016/0014-5793(92)80416-e.]. In it, the author coincidentally refers to the similarity of the chromatin fiber observed in cricket sperm (presumably having a protamine) with that resulting from the nucleosome organization of sea urchin sperm chromatin fibers consisting of histones. This is an important paper that needs to be included in the current manuscript. Of note, the currently abandoned so called chromatin "30 nm fiber" was initially observed and can only be extensively visualized in sperm chromatin of different invertebrate groups.

In the results section and elsewhere in the manuscript, there are several speculative statements that need to be toned down. For instance, on page 5 lines 150-153 the authors state: "Finally, we provide evidence that spooled fibers... an ultracompact configuration that is maintained by disulfide bonding." However, as described by Suzuki and Wakabayashi (reference 40), the cricket sperm consists of a unique sperm nuclear basic protein of approximately 17 kDa for which there is no experimental evidence to date that it contains any cysteine (Analysis of basic proteins during spermatogenesis in the cricket, *Acheta domestica*. Tessier A, Pallotta D. Exp Cell Res. 1973 Nov;82(1):103-10. doi: 10.1016/0014-4827(73)90250-4.). Hence, the reason for the DTT-induced swelling of the sperm nuclei observed in Fig. 6B remains unclear and until a protamine sequence is obtained for this species, it is difficult to attribute it to the cricket SNBP. While a comparison is made to *Drosophila* (fruit fly) sperm for which the cysteine-containing protein sequence of its protamines is well documented, this is not the case with the cricket. The section on histone acetylation is also of a speculative nature and needs to be experimentally developed further before some of the statements made in the manuscript can be made. On page 15, lines 476-478. "The nucleation of acetylation could be driven by a limited number of restricted cis-regulatory DNA segments capable of recruiting HATs ..." again this remains to be proven.

While I do not have any major problem with the model proposed, the discussion section should more clearly emphasize how this model departs from the model initially proposed by Suzuki and Wakabayashi (ref 40 in the manuscript) and where it does not. Also, it would be good if the authors could frame their proposed liquid-crystalline sperm chromatin behaviour within the larger context of liquid-liquid phase separation (LLPS). It would also be good to discuss the potential role of the nucleoplasm forces exerted on to the polymeric (chromatin) fibers [see for instance: [(The Interchromatin Compartment Participates in the Structural and Functional Organization of the Cell Nucleus. Cremer T, Cremer M, Hübner B, Silahtaroglu A, Hendzel M, Lanctôt C, Strickfaden H, Cremer C. Bioessays. 2020 Feb;42(2):e1900132. doi: 10.1002/bies.201900132) and more recently: (Symmetry-based classification of forces driving chromatin dynamics. Eshghi I, Zidovska A, Grosberg AY. Soft Matter. 2022 Oct 14. doi: 10.1039/d2sm00840h)].

MINOR COMMENTS.

1) Page 4, Line 114, a recent review from Rafael Oilva's lab could be included to the references: (Insights into the sperm chromatin and implications for male infertility from a protein perspective. de la Iglesia A, Jodar M, Oliva R, Castillo J. WIREs Mech Dis. 2022 Oct 1:e1588. doi: 10.1002/wsbm.1588.).

- 2) Page 5, line 125, "...vary between species." A reference needs to be added at the end of this sentence: (Spermiogenic chromatin condensation patterning in several hexapods may involve phase separation dynamics by spinodal decomposition or microemulsion inversion (nucleation). Kasinsky HE, Gowen BE, Ausió J. *Tissue Cell*. 2021 Dec;73:101648. doi: 10.1016/j.tice.2021.101648.).
- 3) The molecular mass corresponding to the different bands in the ladder marker need to be indicated.
- 4) Fig. 4.D. EM, the stage of spermiogenesis of the image corresponding transversal section of the nucleus needs to be identified.

Reviewer #2 (Remarks to the Author):

"Biophysical ordering transitions underlie genome 3D re-organization during cricket spermiogenesis." by Orsi et al. describe structural transitions of the genome during spermatogenesis.

The main claim of the manuscript is that polymer stiffness alone, under confinement, can produce structural transitions into an ordered phase. This claim is not new, as shown in some of the referenced work (64, 65) and also perhaps worth mentioning, Reith et al. (<https://doi.org/10.1093/nar/gks157>), Lappala et al. (<https://pubs.acs.org/doi/abs/10.1021/ma4009127>) and most importantly, the most relevant article to this manuscript, a review by Kurt Binder (<https://www.mdpi.com/2073-4360/8/8/296/htm>). Perhaps by addressing some of the results in this paper from the angle of your manuscript would be of benefit.

The article is very visually pleasing, but unfortunately, the beautiful images are not backed by analysis sufficiently. It would be important to quantify, for both the simulations and the experiments, the orientational order parameter, for example, and other properties such as monomer density profiles across the structure for varying values of the stiffness parameter. This manuscript, unfortunately, does not discuss the properties of the system in the nematically ordered phase further than comparing microscopy images to simulation visualizations.

I would like to see comparisons of physical properties, such as the local orientational order parameter, and how this orientational order parameter changes as some other parameters (such as stiffness, twist etc) are varied. To understand this better, and to be able to compare simulation and experiment, one would also have to compute the order parameter for images, and produce statistical interpretations from ensembles of images and simulations, to rule out the possibility of statistically insignificant findings.

We would like to warmly thank the reviewers for taking the time to work on our manuscript and providing us with constructive feedback. We believe we have been able to address all of their concerns, and provide a detailed list of our proposed changes below. Text changes in the revised version related to reviewer's comments are highlighted in red.

REVIEWER COMMENTS

Reviewer #1 (Remarks to the Author):

This is a very interesting paper about the transitions undergone by chromatin during cricket spermiogenesis. In it, a biophysical model simulation based on polymer behavior is developed to account for the chromatin organization transitions observed in combination with electron, confocal and single molecule localization microscopy. The paper can certainly be considered for publication after some extensive rewriting.

We thank the reviewer for their positive appreciation of our initial manuscript. We have abundantly amended our text to account for all of their comments.

COMMENTS: Chromatin modelling using polymer thermodynamic approaches had been previously used to account for the 25-30 nm diameter common to nucleoprotamine and nucleohistone chromatin organization regardless of their non-nucleosome or nucleosome internal organization. [Order and disorder in 30 nm chromatin fibers. Subirana JA. FEBS Lett. 1992 May 11;302(2):105-7. doi: 10.1016/0014-5793(92)80416-e.]. In it, the author coincidentally refers to the similarity of the chromatin fiber observed in cricket sperm (presumably having a protamine) with that resulting from the nucleosome organization of sea urchin sperm chromatin fibers consisting of histones. This is an important paper that needs to be included in the current manuscript. Of note, the currently abandoned so called chromatin "30 nm fiber" was initially observed and can only be extensively visualized in sperm chromatin of different invertebrate groups.

The issue of how the ~25nm fiber is formed in crickets and more generally in other species and cell types, in the non-nucleosomal or nucleosomal chromatin is indeed fascinating. While we did not address polymer formation at this scale in our study, primarily because the cricket TPLs and PLs, if any, remain to be identified, we now discuss this issue and the potential importance of crickets as models to investigate it (lines 474-490).

In the results section and elsewhere in the manuscript, there are several speculative statements that need to be toned down. For instance, on page 5 lines 150-153 the authors state: "Finally, we provide evidence that spooled fibers... an ultracompact configuration that is maintained by disulfide bonding." However, as described by Suzuki and Wakabayasi (reference 40), the cricket sperm consists of a unique sperm nuclear basic protein of approximately 17 kDa for which there is no experimental evidence to date that it contains any cysteine (Analysis of basic proteins during spermatogenesis in

the cricket, *Acheta domestica*. Tessier A, Pallotta D. *Exp Cell Res.* 1973 Nov;82(1):103-10. doi: 10.1016/0014-4827(73)90250-4.). Hence, the reason for the DTT-induced swelling of the sperm nuclei observed in Fig. 6B remains unclear and until a protamine sequence is obtained for this species, it is difficult to attribute it to the cricket SNBP. While a comparison is made to *Drosophila* (fruit fly) sperm for which the cysteine-containing protein sequence of its protamines is well documented, this is not the case with the cricket. The section on histone acetylation is also of a speculative and needs to be experimentally developed further before some of the statements made in the manuscript can be made. On page 15, lines 476-478. "The nucleation of acetylation could be driven by a limited number of restricted cis-regulatory DNA segments capable of recruiting HATs ..." again this remains to be proven.

The reviewer is correct in pointing out that these statements were speculative. We have clarified our wording to avoid any confusion (lines 153-155; 522-527; 556-560).

While I do not have any major problem with the model proposed, the discussion section should more clearly emphasize how this model departs from the model initially proposed by Suzuki and Wakabayashi (ref 40 in the manuscript) and where it does not.

Indeed, Suzuki and Wakabayashi first observed these remarkable bundled chromatin fibers in cricket canoe-stage sperm. We fully validated these observations and provide a complete sequential picture on the dramatic chromatin rearrangements taking place throughout cricket spermiogenesis, and further provide mechanistic insight into how these transitions may operate, exploiting polymer modeling. We have clarified this point in the discussion (lines 465-473).

Also, it would be good if the authors could frame their proposed liquid-crystalline sperm chromatin behaviour within the larger context of liquid-liquid phase separation (LLPS).

We would like to point out that the ordering phase transition that we describe in the paper (in relation with the notion of liquid-crystal) is of a different physical nature than that of LLPS or of micro-phase separation (more appropriate to describe polymer compartmentalization). The former is mainly driven by entropy while the latter by energy.

However, it is true that the LLPS and related notions (spinodal decomposition, micro-emulsion) have been previously hypothesized (eg, Kasinsky HE, Gowen BE, Ausió J. *Tissue Cell.* 2021) as a potential mechanism to drive the formation of the DNA-protamine complexes since the EM images of sperm nuclei in several species may show similarities with the classical patterns observed in arrested spinodal decomposition, phase inversion and emulsion bilayers. We now mention this point in the Introduction (lines 125-130).

It would also be good to discuss the potential role of the nucleoplasm forces exerted on to the polymeric (chromatin) fibers [see for instance: [(The Interchromatin Compartment Participates in the Structural and Functional Organization of the Cell Nucleus. Cremer T, Cremer M, Hübner B, Silahatoglu A, Hendzel M, Lanctôt C, Strickfaden H, Cremer C. *Bioessays.* 2020 Feb;42(2):e1900132. doi: 10.1002/bies.201900132) and more recently: (Symmetry-based classification of forces

driving chromatin dynamics. Eshghi I, Zidovska A, Grosberg AY. *Soft Matter*. 2022 Oct 14. doi: 10.1039/d2sm00840h)].

The impact of nucleoplasm forces on the dynamics of chromatin fiber is a major question. But the importance of such forces is currently still speculative and very few (theoretical and experimental) studies have quantitatively addressed it. The vast majority (if not all) of polymer models investigating chromosome organization do not account explicitly for such forces. We now include a mention of this limitation to clarify our methodology and cite the corresponding papers (lines 722-724 & 743-746).

MINOR COMMENTS.

1) Page 4, Line 114, a recent review from Rafael Oilva's lab could be included to the references: (Insights into the sperm chromatin and implications for male infertility from a protein perspective. de la Iglesia A, Jodar M, Oliva R, Castillo J. *WIREs Mech Dis*. 2022 Oct1:e1588. doi: 10.1002/wsbm.1588.)

The reference has been added as suggested.

2) Page 5, line 125, "...vary between species." A reference needs to be added at the end of this sentence: (Spermiogenic chromatin condensation patterning in several hexapods may involve phase separation dynamics by spinodal decomposition or microemulsion inversion (nucleation). Kasinsky HE, Gowen BE, Ausió J. *Tissue Cell*. 2021 Dec;73:101648. doi: 10.1016/j.tice.2021.101648.).

We thank the reviewer for pointing out this mistake: we did mean to cite this interesting paper originally. It is now cited in the introduction, as suggested, and further highlighted for its interpretations related to LLPS mechanisms (see above).

3) The molecular mass corresponding to the different bands in the ladder marker need to be indicated.

The molecular masses are now indicated.

4) Fig. 4.D. EM, the stage of spermiogenesis of the image corresponding transversal section of the nucleus needs to be identified.

This image corresponds to an elongating spermatid, this information has been added in the legend (now Figure 5D).

Reviewer #2 (Remarks to the Author):

"Biophysical ordering transitions underlie genome 3D re-organization during cricket spermiogenesis." by Orsi et al. describe structural transitions of the genome during spermatogenesis.

The main claim of the manuscript is that polymer stiffness alone, under confinement, can produce structural transitions into an ordered phase. This claim is not new, as shown in some of the referenced work (64, 65) and also perhaps worth mentioning,

Reith et al. (<https://doi.org/10.1093/nar/gks157>), Lappala et al. (<https://pubs.acs.org/doi/abs/10.1021/ma4009127>) and most importantly, the most relevant article to this manuscript, a review by Kurt Binder (<https://www.mdpi.com/2073-4360/8/8/296/htm>). Perhaps by addressing some of the results in this paper from the angle of your manuscript would be of benefit.

We thank the reviewer for their constructive comments. The reviewer is correct in pointing out that the notion that polymer stiffness results in nematic ordering is not new, and we do not claim that this is an original result of our present manuscript. Instead, we would like to stress that the novelty of our study lies in the complete characterization of a unique large-scale genome rearrangement in cricket sperm as a twisted spool, which has to our knowledge never been described before *in vivo*. Furthermore, another central finding of our study is that the spontaneous establishment of this ordered structure starting from flexible, disordered polymer coil cannot be achieved by a simple, uniform increase of the fiber bending stiffness, but rather requires nucleated and progressive fiber rigidification kinetics, which are fully consistent with our *in vivo* observations. Following the reviewer's suggestions, in order to better highlight the novelty of our conclusions in the context of current literature, we have now extended this part of the discussion, including the indicated references (lines 573-583).

The article is very visually pleasing, but unfortunately, the beautiful images are not backed by analysis sufficiently. It would be important to quantify, for both the simulations and the experiments, the orientational order parameter, for example, and other properties such as monomer density profiles across the structure for varying values of the stiffness parameter. This manuscript, unfortunately, does not discuss the properties of the system in the nematically ordered phase further than comparing microscopy images to simulation visualizations.

I would like to see comparisons of physical properties, such as the local orientational order parameter, and how this orientational order parameter changes as some other parameters (such as stiffness, twist etc) are varied. To understand this better, and to be able to compare simulation and experiment, one would also have to compute the order parameter for images, and produce statistical interpretations from ensembles of images and simulations, to rule out the possibility of statistically insignificant findings.

We thank the reviewer for suggesting these analyses, which we now provide and greatly reinforce our findings. In particular, we focused on the quantifications of positional and orientational properties of the system as a function of the model parameters such as fiber stiffness, nuclear extension force and detailed rigidification mechanism.

Specifically:

- We now present a new analysis showing how the local orientational order parameter, the surface nematic order parameter, and the density parameter change as persistence length is tuned in our global and nucleated rigidification simulations, and as nuclear stretching is implemented (new Figures 2E, 3C, 3D, 5E and Supplementary Figures 2A, 2B and 3).

- As suggested, we further calculated the orientational order parameter from ensembles of electron microscopy images of spermatocytes (i.e before chromatin spooling), and of

canoe-stage spermatids (i.e in spooled genome)(reported in new Figures 3D, 5E and Supplementary Figure 3), accounting for their variability.

- Also as suggested, we further measured the radial density variations of the nuclear content by exploiting transversal sections of spermatocytes, round spermatids, and elongating spermatids (new Figure 2F), and compared this to the monomer density in our simulations at different persistence lengths using various fiber rigidification pathways (new Figure 2E and 3C).

These analyses identify a plausible range of persistence lengths and stretching forces in our “nucleated rigidification” model that are compatible with the typical orders of magnitudes reported in other biological systems, further supporting the validity of our model.

- Finally, we have included a supplementary figure with additional representative source images used for these analyses (new Supplementary Figure 1), to further illustrate that the observed genome organization transitions are systematic in cricket sperm.

Altogether, we have substantially revised our manuscript according to the reviewer’s suggestions, adding in this new version 5 new main Figure panels, leading to a total of 7 main Figures (compared to 6 in the first version), 3 new supplementary figures, as well as over 3 pages of new/revised text, including 10 additional references. We hope that you agree with our view that all of these additions consolidate and strengthen our original findings and claims.

REVIEWERS' COMMENTS

Reviewer #2 (Remarks to the Author):

Thank you for the changes you implemented with care and rigor. I am pleased with the new analysis and think that the paper really benefitted from these measurements.